

# Vegetation optimality explains the convergence of catchments on the Budyko curve

Remko C. Nijzink[1] and Stanislaus J. Schymanski[1]

[1]Catchment and Ecohydrology Group (CAT), Environmental Research and Innovation (ERIN), Luxembourg Institute of Science and Technology (LIST), Belvaux, Luxembourg

**Correspondence:** remko.nijzink@list.lu

**Abstract.**

The Budyko framework puts the long-term mean annual evapo-transpiration (ET) of a catchment in relation to its maximum possible value determined by the conservation of mass (ET can not exceed mean annual precipitation) and energy (ET can not exceed mean annual net radiation) in the absence of significant storage contributions. Most catchments plot relatively close to this physical limit, which allowed to develop an empirical equation (often referred to as the Budyko curve) for estimating mean annual evaporation and runoff from observed net radiation and precipitation. Parametric forms of the curve often use a shape parameter ($n$), that is seen as a catchment characteristic. However, a satisfying explanation for the convergence and self-organization of catchments around such an empirical curve is still lacking. In this study, we explore if vegetation optimality can explain the convergence of catchments along a Budyko curve and in how far $n$ can be seen as a catchment characteristic.

The Vegetation Optimality Model (VOM) optimizes vegetation properties and behaviour (e.g. rooting depths, vegetation cover, stomatal control), to maximize the difference between the total carbon taken up from the atmosphere and the carbon used for maintenance of plant tissues involved in its uptake, i.e. the long-term net carbon profit (NCP). This optimization is entirely independent of observed ET and hence the VOM does not require calibration for predicting ET. In a first step, the VOM was fully optimized for the observed atmospheric forcing at five flux tower sites along the North Australian Tropical Transect, as well as 36 additional locations near the transect and six Australian catchments. In addition, the VOM was run without vegetation for all sites, meaning that all precipitation was partitioned into soil evaporation and runoff. For comparison, three conceptual hydrological models (TUWmodel, GR4J and FLEX) were calibrated for the Australian catchments using the observed precipitation and runoff. Subsequently, we emulated step changes in climate by multiplying precipitation ($P$) by factors ranging between 0.2 and 2, before running the VOM and hydrological models without changing the vegetation properties or model parameters, emulating invariant catchment characteristics under a changed climate. In a last step, the VOM was re-optimized for the different $P$ amounts, allowing vegetation to adapt to the new situation. Eventually, Budyko curves were fit by adapting the parameter $n$ to the model results. This was carried out for both multiple sites simultaneously and for each individual study site, thereby assuming that $n$ is a site specific characteristic.

The optimized VOM runs tracked relatively close to a Budyko curve with a realistic $n$ value and close to observations, whereas the runs without vegetation led to significantly lower evaporative fractions and unrealistically low $n$ values compared with literature. When fitting $n$ to individual catchments, changes in $P$ led to changes in $n$ (increasing $n$ for decreasing $P$) in all





model runs (including the three conceptual models) except if the VOM was re-optimized for each change in $P$, which brought the value of $n$ back close to its value for the unperturbed $P$ in each catchment. For the re-optimized VOM runs, the variation in $n$ between catchments was greater than within each catchment in response to multiplications of $P$ with a factor 0.2 to 2.

These findings suggest that optimality may explain the self-organization of catchments in Budyko space, and that the ac-
companying parameter $n$ does not remain constant for constant catchment and vegetation conditions as hypothesized in the literature, but in fact emerges through the adaptation of vegetation to climatic conditions in a given hydrological setting. More-over, the results suggest that $n$ might initially increase in response to suddenly reduced $P$, and only slowly returns to its original, catchment-specific value, as vegetation re-adjusts to the new climate over decades and centuries. This may constitute a new basis for the evaluation and prediction of catchment responses to climatic shifts.

## 1 Introduction

Conservation of mass is a fundamental physical law that is commonly invoked in catchment hydrology. Essentially, it means that the net exchange of water across catchment boundaries (due to precipitation ($P$), evapo-transpiration ($E_T$) and discharge ($Q$)) equals the change in catchment water storage over time. Another fundamental physical law is conservation of energy,
which puts an upper bound on how much water can leave the catchment by evapo-transpiration, as the phase change from liquid to gaseous water requires a large amount of energy, which is mainly supplied by the sun. When considering long time scales (e.g. decades), a potential change in storage becomes tiny compared to the cumulative fluxes, which allows the prediction that the total evapo-transpiration ($E_T$) of a catchment can not exceed the integrated amount of precipitation over the same time period. Another limit is defined by the integrated amount of energy available for $E_T$ ("potential evaporation", $E_p$), which the
total evapo-transpiration ($E_T$) can not exceed either. Already in the early 1900s, scientists found out that $E_T$ estimated from the difference between long-term precipitation and runoff ($E_T = P - Q$) tracks relatively close to these limits in large catchments, and they proposed empirical equations to predict mean annual $E_T$ and runoff from observed net radiation and precipitation (Ol'Dekop, 1911; Schreiber, 1904). Budyko (1974) combined these empirical equations to what we call here the Budyko curve and formulated a framework with a catchment dryness index $E_p/P$ as independent variable and the evapo-transpired fraction
of precipitation ($E_T/P$) as dependent variable. Note that the framework can alternatively be presented with a wetness index $P/E_p$ as independent variable (Yang et al., 2008; Roderick and Farquhar, 2011), as introduced by Pike (1964).

The Budyko framework has been widely used in catchment science, and has proven to be a powerful tool in order to assess the water balance in relation to its physical boundaries. Early focus fell on explaining the spread around the empirical curve. For example, Budyko (1974) argued that downwards deviations from the curve are bigger where seasonal signals of potential
evaporation and rainfall were out of phase, which was later confirmed by others (Yokoo et al., 2008; Potter et al., 2005). It was





also shown that variations in soil moisture storage capacity (e.g. Milly, 1994; Woods, 2003), land cover (Oudin et al., 2008) and vegetation (Donohue et al., 2007) are able to explain deviations from the curve.

As more catchments were analysed in the Budyko framework, more systematic deviations from the original Budyko curve were discovered, motivating more flexible formulations with an additional parameter to adjust the shape of the curve (Fu, 1981; Choudhury, 1999; Zhang et al., 2001, 2004). Yang et al. (2008) demonstrated that the two widespread flexible parameter

formulations of Fu (1981) and Choudhury (1999) (also attributed to Mezentsev (1955)) are approximately equivalent if their shape parameters followed a certain linear relationship to each other. Therefore, we will use the formulation by Choudhury (1999) and refer to the parameter as $n$ here, regardless of its name in the different publications cited hereafter. Interestingly, the meaning of the additional parameters were explained in different ways by different authors. For example, Zhang et al. (2001) called $n$ the plant available water coefficient, while Ning et al. (2017) fitted separate values of $n$ each year and related these

values to annual measures of seasonality and vegetation cover. Donohue et al. (2012) used a multi-variate approach to explain variations in $n$ by local rooting depths, storm depths and soil water storage capacities. Similarly, Roderick and Farquhar (2011) argued that this parameter should be considered a result of all local conditions combined (except for climate). Specifically, in their analysis of runoff sensitivity to perturbations in $P$ and $E_p$, they held $n$ constant, followed by a separate sensitivity analysis to perturbations in $n$, arguing that $n$ would only change over longer time scales, e.g. due to a change in vegetation cover.

While explanations for deviations from the Budyko curve (e.g. Roderick and Farquhar, 2011; Donohue et al., 2012; Ning et al., 2017) and the practical use of the Budyko framework (e.g. Nijzink et al., 2018; Hulsman et al., 2018; Mianabadi et al., 2019) is being explored intensively to this day, less attention has been put on explaining why catchments converge on a curve in Budyko space at all, instead of randomly falling somewhere in the envelope determined by the conservation of mass and energy. Optimality theory could provide a promising avenue to explain this convergence, as it allows selecting the most likely states

of a system from the range of possible states. Wang et al. (2015) and Westhoff et al. (2016) used thermodynamic optimality principles (maximum entropy production and maximum power, respectively) to produce curves resembling the Budyko curve, but they did not explain the role of vegetation for the shape of the curve, as suggested by many authors (e.g. Roderick and Farquhar, 2011; Donohue et al., 2007; Oudin et al., 2008; Yang et al., 2009; Williams et al., 2012). Here we investigate if vegetation optimality explains the convergence of catchments on the Budyko curve. Vegetation optimality proposes that

vegetation self-optimizes to maximize its long-term net carbon profit (NCP), which is the difference between carbon taken up during photosynthesis and carbon invested in plant organs involved in carbon and water uptake, and transport. The principle was implemented by Schymanski et al. (2009) in the Vegetation Optimality Model (VOM), which couples a vegetation and water balance model, and was shown to reproduce observed carbon and water fluxes at several tropical savanna sites in Australia (Schymanski et al., 2009; Nijzink et al., 2022a) and explain general trends in vegetation responses to elevated atmospheric

$CO_2$ concentrations (Schymanski et al., 2015). The model distinguishes between slowly optimizing vegetation properties, such as water use strategies, tree cover and rooting depths, and quickly optimizing vegetation properties, such as photosynthetic capacities, vertical root distributions, grass cover and stomatal conductances. The former are held constant over decades, while the latter vary at a seasonal or even hourly scale. This enables the distinction between day-to-day responses of vegetation to





environmental drivers, and the slow responses (e.g. tree cover, species composition) assumed to result in changing $n$-values (Roderick and Farquhar, 2011).

Based on the above considerations, we formulated the following hypotheses:

H1 Model simulations based on vegetation optimality lead to a better reproduction of the empirical Budyko curve than model simulations without self-optimized vegetation.

5  H2 The empirical parameter $n$ stays constant as climate changes, as long as vegetation cover and rooting depths stay constant.

H3 Changes in $n$-values are a result of slowly varying, long-term vegetation properties.

## 2 Methodology

In order to address the hypotheses, three hydrological models and the VOM were applied to several sites in Australia. All the analyses and model runs were carried out with an open science approach by using the platform RENKU (https://renkulab.io/, 10  last access: 10 February 2022), that tracks all steps in the scientific process. The resulting work flows including code and data can be found online (https://renkulab.io/projects/remko.nijzink/budyko, last access: 10 February 2022).

### 2.1 Budyko formulations

In this study, we start from Eq. 1 in Roderick and Farquhar (2011) (which is equivalent to Eq. 3 in Choudhury (1999) and was traced back to Mezentsev (1955) by Yang et al. (2008)):

$$15 \quad \overline{E_a} = \frac{\overline{E_p P}}{\left(\overline{P}^n + \overline{E_p}^{-n}\right)^{1/n}} \tag{1}$$

with $\overline{E_p}$ the mean annual potential evaporation, $\overline{E_a}$ the mean annual evaporation, $\overline{P}$ the mean annual precipitation, and $n$ a shape factor, assumed to represent catchment characteristics (e.g. vegetation, soils). To express the ratio $\overline{E_a}/\overline{P}$ as a function of $\overline{E_p}/\overline{P}$ (see also Supplement S5), both sides of Eq. 1 were divided by $\overline{P}$ and the resulting nominator and denominator on the right hand side were again divided by $\overline{P}$. Following Budyko (1974), $\overline{E_p}$ was expressed as a function of mean annual net 20  radiation ($\lambda \overline{E_p} = \overline{R_n}$), and all fluxes were transformed in terms of energy by multiplying by the latent heat of vaporization $\lambda$:

$$\frac{\lambda \overline{E_a}}{\lambda \overline{P}} = \frac{\overline{R_n}}{\lambda \overline{P}} \left(\left(\frac{\overline{R_n}}{\lambda \overline{P}}\right)^n + 1\right)^{-1/n} \tag{2}$$



In a similar way, Eq. 1 can be transformed into an equation with the fraction of $\lambda\overline{P}/\overline{R_n}$ as the independent variable (see also Supplement S5), by first dividing both sides of Eq. 1 by $\overline{E_p}$, followed by a division of the nominator and denominator of the right hand side by $\overline{E_p}$ and a conversion to energetic units:

$$\frac{\lambda\overline{E_a}}{\overline{R_n}} = \frac{\lambda\overline{P}}{\overline{R_n}}\left(\left(\frac{\lambda\overline{P}}{\overline{R_n}}\right)^n + 1\right)^{-1/n} \tag{3}$$

For fits to single data points (i.e. one catchment/study site) the equations were solved analytically using open source software (python sympy, https://www.sympy.org/, last access: 10 February 2022, python essm https://pypi.org/project/essm/, last access: 10 February 2022). The exponent $n$ in Equations 2 and 3 was fitted to data of multiple catchments with a non-linear least squares fit based on the Levenberg-Marquard algorithm (python scipy.optimize.curve_fit, https://docs.scipy.org/doc/scipy/reference/generated/scipy.optimize.curve_fit.html, last access: 10 February 2022, Levenberg, 1944). The mean squared error (MSE) was used in order to assess the goodness of fit:

$$MSE = \frac{1}{m}\sum_{i=1}^{m}\left(Y_i - \overline{Y}_i\right)^2 \tag{4}$$

with $m$ the number of observations, $Y_i$ an individual observation and $\overline{Y}_i$ the predicted value.

## 2.2 Study sites

In order to capture a variation of climates, the study focuses on several sites along a precipitation gradient in Australia. The study sites include flux tower sites where the VOM has been tested previously (Nijzink et al., 2022b, a), as well as several larger catchments. An additional analysis based on a selection of 357 catchments of the CAMELS dataset (Addor et al., 2017; Newman et al., 2015) is presented in Supplement S3, Figures S3.8-S3.11.

### 2.2.1 North Australian Tropical Transect

The Vegetation Optimality Model (VOM, see Section 2.3) was set up for five flux tower sites that are located between $12.5°S$ and $22.5°S$ along the North Australian Tropical Transect (NATT, Hutley et al., 2011). The precipitation decreases from 1700 to 500 mm/year from north to south along the transect, over a distance of approximately 1000 km. The five sites used in this study are summarized in Table 1 and the geographical locations are shown in Figure 1, see also Nijzink et al. (2022b) and Nijzink et al. (2022a). A more detailed description of the sites can be found in Hutley et al. (2011). In addition to the flux tower sites, 36 locations along the transect were used (see also Figure 1), which were all located between $12.5°S$ and $17.5°S$, and $131.0°E$ and $133.5°S$, with a distance between the locations of $0.5°$.

The soil parameterization of the VOM is based on data of the Soil and Landscape Grid of Australia (Viscarra Rossel et al., 2014a, b, c), in addition to specific site descriptions provided by Hutley et al. (2011) and Whitley et al. (2016). For a description of the soil profiles, see also Supplement S8 of Nijzink et al. (2022a). Meteorological data from the Australian SILO Data Drill

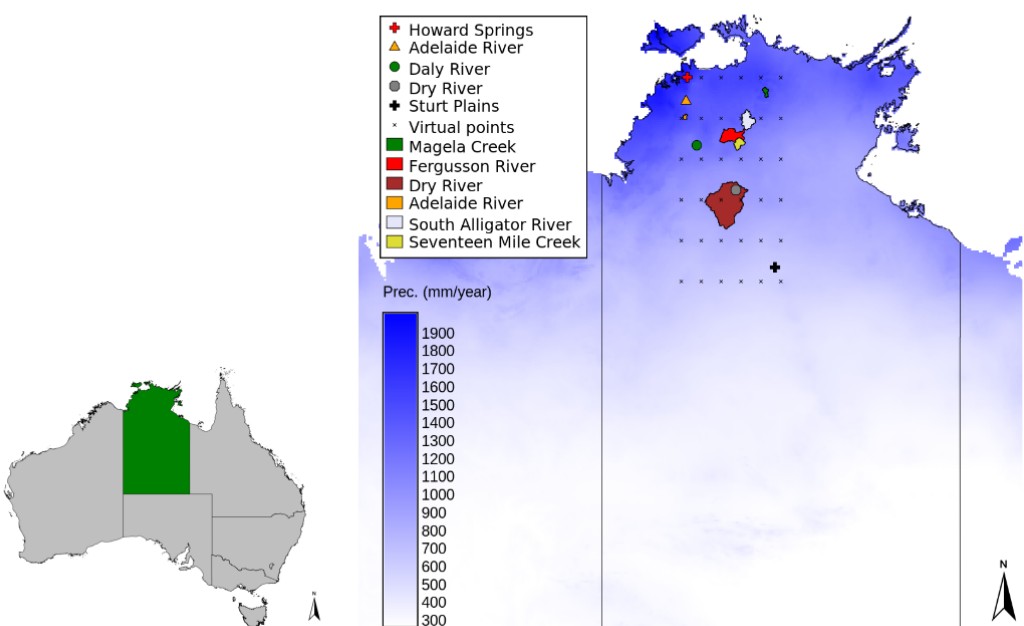

**Figure 1.** Catchments (polygons) and flux tower sites along the Nort Australian Tropical Transect (symbols), with additional locations shown as crosses. The mean annual precipitation is indicated by the blue colorscale

(Jeffrey et al., 2001) was used to run the model and consisted of time series of daily maximum and minimum temperatures, shortwave radiation, precipitation, vapour pressure and atmospheric pressure. The Mauna Loa $CO_2$-records (Keeling et al., 2005) provided time series of atmospheric $CO_2$-concentrations.

### 2.2.2 Australian catchments

The NATT sites described above were not part of any hydrologically gauged catchments and, therefore, we selected an additional six catchments with runoff data, which were close to the NATT sites and had similar climates. This allowed to compare VOM results with the results of three relatively simple, conceptual hydrological models (FLEX, TUWmodel and GR4J, see

5 Section 2.4), which require runoff data for calibration. These catchments were also previously used by Zhang et al. (2004) in their Budyko-analysis. See Table 2 for details of these catchments and Figure 1 for their locations. Meteorological data was again taken from the Australian SILO Data Drill (Jeffrey et al., 2001), from which time series of potential evaporation (FAO, Penman-Monteith formula, Allen et al., 1998), precipitation and daily maximum and minimum temperatures were used to run





**Table 1.** Characteristics of the study sites along the North Australian Tropical Transect, vegetation data from Hutley et al. (2011) and Whitley et al. (2016), with Eucalyptus (Eu.), Erythrophleum (Er.), Terminalia (Te.), Corymbia (Co.), Planchonia (Pl.), Themeda (Th.), Hetropogan (He.), and Chrysopogon (Ch.). Meteorological data is taken from the SILO Data Drill (Jeffrey et al., 2001) for the model periods of 1-1-1980 until 31-12-2017, including the FAO Penman-Monteith potential evaporation (Allen et al., 1998). Aridity is defined as the ratio of net radiation to precipitation (multiplied by the latent heat of vaporization $\lambda$), $R_n/(\lambda P)$. Tree cover is determined as the minimum value of the mean monthly projective cover based on remotely sensed observations of the fraction of photosynthetically active radiation (fPAR Donohue et al., 2008). The maximum grass cover was found by subtracting the tree cover from the remotely sensed projective cover.

| Study Site | Howard Springs | Adelaide River | Daly River | Dry River | Sturt Plains |
|---|---|---|---|---|---|
| FLUXNET ID | AU-How | AU-Ade | AU-DaS | AU-Dry | AU-Stp |
| Coordinates | 12.49S | 13.08S | 14.16S | 15.26S | 17.15S |
| | 131.35E | 131.12E | 131.39E | 132.37E | 133.35E |
| Prec. (mm year$^{-1}$) | 1747 | 1497 | 1166 | 898 | 616 |
| Pot. evap. (mm year$^{-1}$) | 1763 | 1802 | 1896 | 1948 | 2082 |
| Aridity. (-) | 1.03 | 1.18 | 1.48 | 1.87 | 2.70 |
| Net Rad. (MJ m$^{-2}$ year$^{-1-1}$) | 4392 | 4313 | 4215 | 4105 | 4079 |
| Mean max. temp. [$^o$C] | 37.5 | 38.8 | 40.6 | 41.1 | 43.0 |
| Mean min. temp. [$^o$C] | 27.4 | 26.6 | 26.9 | 27.7 | 28.1 |
| Tree cover (%) | 39.8 | 20.8 | 37.5 | 26.6 | 7.4 |
| Max. grass cover (%) | 44.3 | 59.2 | 42.5 | 49.4 | 57.6 |
| Species | | | | | |
| Overstorey | *Eu. miniata* | *Eu. tectifica* | *Te. grandiflora* | *Eu. tetrodonta* | - |
| | *Eu. tetrodonta* | *Co. latifolia* | *Eu. tetrodonta* | *Co. terminalis* | |
| | *Er. chlorostachys* | *Pl. careya* | *Co. latifolia* | *Eu. dichromophloia* | |
| Understorey | *Sorghum spp.* | *Sorghum spp.* | *Sorghum spp.* | *Sorghum intrans* | *Astrebla spp.* |
| | *He. triticeus* | *Ch. fallax* | *He. triticeus* | *Th. Tiandra* | |
| | | | | *Ch. fallax* | |

the hydrological models. The VOM used again time series of the SILO Data Drill of daily maximum and minimum temperatures, shortwave radiation, precipitation, vapour pressure and atmospheric pressure. The Mauna Loa $CO_2$-records (Keeling et al., 2005) provided again time series of atmospheric $CO_2$-concentrations for the VOM-simulations.

## 2.3 Vegetation Optimality Model

The Vegetation Optimality Model (VOM, Schymanski et al., 2009, 2015; Nijzink et al., 2022b) is a combined water and vegetation model, that optimizes vegetation properties, such as rooting depths and foliage cover, in order to maximize the





**Table 2.** Characteristics of the six Australian catchments.

| Study Site | Area (km²) | Aridity (-) | Prec. (mm year⁻¹) | Pot. Evap. (mm year⁻¹) | Discharge (mm year⁻¹) | Data availability |
|---|---|---|---|---|---|---|
| Adelaide River | 115.0 | 1.17 | 1469 | 1715 | 282 | 27-08-1981 – 13-07-2020 |
| Dry River | 8235.9 | 2.24 | 848 | 1895 | 27 | 01-01-1980 – 13-07-2020 |
| Fergusson River | 1686.1 | 1.51 | 1160 | 1750 | 327 | 01-01-1980 – 19-04-2020 |
| Magela Creek | 265.5 | 1.08 | 1457 | 1580 | 695 | 01-01-1980 – 02-11-2006 |
| Seventeen Mile Creek | 602.2 | 1.55 | 1133 | 1756 | 363 | 01-01-1980 – 13-07-2020 |
| South Alligator River | 1218.0 | 1.31 | 1267 | 1663 | 505 | 01-01-1980 – 08-06-2010 |

net carbon profit (NCP), defined here as the difference between carbon taken up by photosynthesis and the carbon invested into maintenance of leaves, roots and water transport tissues. The model code and documentation can be found online (https://github.com/schymans/VOM, last access: 10 February 2022, https://vom.readthedocs.io, last access: 10 February 2022) and a more detailed description can be found in Schymanski et al. (2009, 2015) and Nijzink et al. (2022b). VOM version v0.6 (https://github.com/schymans/VOM/tree/v0.6, last access: 4 March 2022) was used in this study.

### 2.3.1 Vegetation model

Vegetation is schematized in the VOM as two big leaves, with one leaf representing the perennial vegetation (trees) and one leaf representing the annual grasses. The photosynthesis of these leaves is calculated based on a simplified canopy-gas exchange model for C3 plants (Schymanski et al., 2007), based on von Caemmerer (2000), which uses irradiance, atmospheric $CO_2$ concentration, temperature, photosynthetic capacity and stomatal conductance. Stomatal conductance and observed water vapour deficit are used to compute transpiration rates, while root water uptake is driven by the water potential difference between the plant and each soil layer, following an electrical circuit analogy (Schymanski et al., 2008).

In order to calculate the NCP, the VOM subtracts respiration of roots and leaves as well as the maintenance and turnover carbon costs of foliage from the photosynthetic carbon uptake. In addition, carbon costs for the water transport system are represented as a function of rooting depth and projected vegetation cover and also subtracted from the carbon uptake.

Each optimized vegetation property thus incurs both a benefit (e.g. increasing light or water supply for photosynthesis) and a cost (construction/maintenance), defining the NCP, but they are optimized at different time scales. Rooting depths of the perennial trees and seasonal grasses ($y_{r,p}$ and $y_{r,s}$, respectively) as well as the foliage projected cover of the perennial vegetation ($M_{A,p}$) and two parameters defining the water use strategy of each big leaf are assumed to be constant during the simulation period of 37 years and optimized to maximize the NCP over the entire period. In contrast, seasonal vegetation cover and photosynthetic capacities of the seasonal and perennial vegetation are adjusted incrementally from day to day based on the daily NCP on the previous day. At the same time, as soil moisture gets depleted, soil water potential declines and if canopy





water demand cannot be matched by root water uptake, the model reduces stomatal conductance below its optimized value and increases root surface area.

### 2.3.2 Water balance model

The water balance part of the model (see also  Schymanski et al., 2008, 2015) was set up as described by Nijzink et al. (2022b). Briefly, it consists of a permeable soil block with layers of 0.2 m thickness, and a total thickness of 30 m, parameterized in
a way to resemble freely draining conditions, which was done due to the absence of detailed knowledge about the hydrology of the sites. Precipitation can either infiltrate, directly run off as surface runoff or evaporate right away as soil evaporation, depending on the saturation of the top soil layer. Afterwards, water can either percolate further down towards more saturated layers, or be taken up by roots for transpiration. A drainage flux sets in as soon as the water table exceeds a prescribed drainage level, which is set here to 25 m below the surface. In this way, the groundwater table was kept well outside the reach of roots,
i.e. resembling free drainage conditions.

### 2.3.3 Optimization

The Shuffled Complex Evolution algorithm (SCE, Duan et al., 1994) was used to optimize the long-term parameters in the VOM for the full simulation period of 37 years (1-1-1980 until 31-12-2017). The number of initial complexes was set to 10, after which the algorithm performs a local optimization within each complex. Afterwards, the complexes are mixed, aiming
to find the global optimum. For the day-to-day optimization of the variable vegetation properties, the model was run with the actual and a higher and lower value of each property for each day, and the properties were adjusted at the end of each day to the combination that would have led to the maximum daily NCP.

## 2.4   Conceptual hydrological models

### 2.4.1   TUWmodel

The TUWmodel (https://cran.r-project.org/web/packages/TUWmodel/, last access: 10 February 2022,  Parajka et al., 2007) is a version of the widely used HBV-model (Bergström, 1976), that consists of a set of reservoirs in series. The effective precipitation is determined by a snow module, after which it enters a soil moisture module. The current soil moisture state determines how much of the effective precipitation will infiltrate and how much will go to a fast reservoir for runoff. The transpiration rate depends on the fractional filling of the soil moisture reservoir as well as the potential evaporation:

$$E_T = \begin{cases} \frac{S_m}{LP} * E_p, & \text{if } S_m < LP \\ E_p, & \text{if } S_m >= LP \end{cases} \qquad (5)$$

with $S_m$ the soil moisture state, $LP$ a soil moisture threshold after which evaporation occurs at a potential rate. Furthermore, the fast reservoir has an overflow outlet, that accounts for the fast overland flow component, and a normal reservoir outlet.



From this fast reservoir, water can also percolate further down, to a slow reservoir that represents the groundwater component. Eventually, a triangular routing function is applied to the runoff components to determine the final discharge. The TUWmodel has 15 parameters that need to be calibrated against streamflow data (See Table S1 in the Supplement for prior parameter ranges).

### 2.4.2 GR4J

5 The GR4J-model (https://cran.r-project.org/web/packages/airGR/, last access: 10 February 2022, Perrin et al., 2003) was used in our study with the additional snow module of Valéry et al. (2014). At first, the snow module determines the amount of melt water and liquid rainfall. If the rainfall and snowmelt exceed the potential evaporation, the net precipitation is determined by subtracting the potential evaporation from the rainfall and snowmelt, and the net available potential evaporation is set to zero. Conversely, if the rainfall and snowmelt are less than the potential evaporation, the net precipitation is set to zero and 10 the net available potential evaporation is determined by subtracting the rainfall and snowmelt from the potential evaporation. Afterwards, a part of the net precipitation enters a reservoir, defined as the production store, based on the current level of storage in this reservoir. The actual evaporation is determined based on the actual levels in this production store:

$$E_t = \frac{S\left(2 - \frac{S}{x_1}\right)\tanh\left(\frac{E_n}{x_1}\right)}{1 + \left(1 - \frac{S}{x_1}\right)\tanh\left(\frac{E_n}{x_1}\right)} \tag{6}$$

with $S$ the actual storage, $x_1$ the maximum storage capacity and $E_n$ the net available potential evaporation. From the 15 production store, water can also percolate down. Eventually, the percolated water is added to the part of the precipitation that did not enter the production store, and this sum of water is divided into two flow components. 90% is routed by a unit hydrograph and a non-linear routing store, whereas 10% is routed by a single unit hydrograph. The two routed components are summed again to obtain the resulting discharge. The GR4J-model with the additional snow module has in total six model parameters for calibration.

### 2.4.3 FLEX

20 The last hydrological model (https://github.com/rcnijzink/flexsimple/, last access: 10 February 2022) used in this study is based on the FLEX model as originally described by Fenicia et al. (2006). At first, a snow module is run to determine the amount of water that enters the interception reservoir. From there, the water either evaporates directly, or, when a storage threshold is exceeded, the water continues to the unsaturated reservoir. Depending on the current state in the unsaturated storage, water 25 can infiltrate into the unsaturated reservoir, or is added to a fast flow reservoir. Evaporation takes place from the unsaturated reservoir as well, based on the amount of water stored here:

$$E_T = \begin{cases} \frac{S_u}{LP * S_{u,max}} * E_p, & \text{if } S_u < LP * S_{u,max} \\ E_p, & \text{if } S_u < LP * S_{u,max} \end{cases} \tag{7}$$





with $S_u$ the soil moisture state, $S_{u,max}$ the maximum soil moisture storage, $LP$ a relative soil moisture threshold after which evaporation occurs at a potential rate. Besides evaporation, percolations occures as well from the unsaturared reservoir, based again on the amount of water in the unsaturated zone. The percolated water adds to the last reservoir, which mimics the slow component of the groundwater. The FLEX model has 10 free parameters for calibration.

### 2.4.4 Calibration strategy for the conceptual hydrological models

The three hydrological models were each run with 50,000 random parameterizations for the six Australian catchments. This was done in order to find the most suitable parameter set, which is defined here as the parameter realization that achieves the highest Kling-Gupta efficiency (KGE, Gupta et al., 2009) for reproducing observed discharge during the calibration period. The full time series are used for the calibration period, after omitting the first warm-up year. The models were run for the full length of the time series as specified in Table 2.

### 2.5 Experimental design

The general approach was to first run the models using site-specific meteorology, and then conduct several numerical experiments where only precipitation is increased or decreased by a constant factor, while all other meteorological variables and model parameters remain unaltered. Precipitation was chosen as a representation of climate change, as it affects all models in a similar way, whereas potential evaporation is not used as input for the VOM. Since the VOM requires extended meteorological input over the entire modelling period (solar irradiance, vapour pressure), whereas the conceptual hydrological models require streamflow data for calibration, they were run for different sets of sites, with only the six catchments in Australia providing adequate input data for all models. For the Australian catchments and flux tower sites, the positions in the Budyko framework for the VOM and the hydrological models were all determined for the period 1-1-1985 until 31-12-2005 for consistency.

In the first step, the VOM was fully optimized for the flux tower sites, the 36 additional locations along the NATT, and the six Australian catchments for the meteorological data as observed. At the same time, the three hydrological models FLEX, TUWmodel and GR4J were calibrated (see sect. 2.4.4) for the six catchments in Australia also with the observed site-specific meteorological data. Supplement S3 contains an additional analysis where this was repeated for 357 catchments in the US, to assess the behaviour of the hydrological models also for a larger set of catchments with different climates and vegetation.

In the next step, the meteorological forcing was altered by multiplying the precipitation ($P$) with factors ranging from 0.2 to 2.0 in steps of 0.2, and the VOM was run for each $P$ scenario with the vegetation parameters that were obtained in the unmodified situation based on observed $P$, representing perturbations in climate where long-term vegetation properties are not (yet) affected by changes in $P$. Besides that, the VOM was also run without vegetation (i.e. only soil evaporation) for the different $P$ amounts. Similarly, the conceptual hydrological models were run for each $P$ scenario as well, using the parameters that were calibrated to the unmodified situation.

In a third step, the VOM was re-optimized for all $P$ scenarios at each of the Australian study sites, representing perturbations in climate where vegetation has fully adapted to each perturbation.





## 3 Results

### 3.1 VOM simulations with unperturbed rainfall

VOM simulations with optimized vegetation properties led to a closer convergence along the Budyko curve and higher $n$ values
than VOM simulations without vegetation (Figure 2). For the flux tower sites where observations of $\lambda E$ were available (Figure

2a), the optimized VOM plotted much closer to the flux tower observations than the VOM-runs without vegetation. At the
same time, the curve fit for Eq. 3 was much better for the optimized VOM, as indicated by the lower mean squared error (3.86
$\cdot 10^{-3}$ vs. $2.004 \cdot 10^{-2}$). The 36 additional locations along the NATT (Figure 2b) and the six Australian catchments (Figure
2c) also resulted in higher $n$ values for the optimized vegetation compared to no vegetation, as well as a higher convergence to
the curve (lower mean squared errors, with $1.788 \cdot 10^{-3}$ vs $2.339 \cdot 10^{-2}$, and $1.249 \cdot 10^{-3}$ vs $5.504 \cdot 10^{-3}$, for the curves with

and without optimized vegetation respectively).

### 3.2 Optimal vegetation response to modified precipitation

Simulated responses to systematic shifts in precipitation ($P$) only tracked along the Budyko curve at individual flux tower sites
if vegetation was re-optimized (i.e green and gray symbols did not plot along the lines of the same colour in Figure 3, but the
black symbols plotted much closer to their curve). This is also evident in the substantially lower mean squared errors for the

black lines (see figure legends). The deviations from the Budyko curves were systematic, in that simulations with reduced $P$
(higher $Rn/\lambda P$) fell above the curve and simulations with higher $P$ (lower $Rn/\lambda P$) fell below the curve.

The $n$-value fitted for 36 additional locations along the NATT similarly reduced from 1.183 to 1.095 after increasing the
total rainfall by 20% if the long-term vegetation properties were not re-optimized (Figure 4). After re-optimizing the vegetation
properties in the VOM for the increased precipitation, $n$ returned close to its original value for unperturbed precipitation (black

triangles in Figure 4, with an $n$-value of 1.176). This was associated with increased values of the water use parameters ($c_{\lambda f,p}$ and
$c_{\lambda e,p}$), vegetation cover and rooting depth of the perennial vegetation (Figure 5a), whereas the seasonal vegetation parameters
changed less. Re-optimization for increased $P$ also resulted in greater water and carbon fluxes in the VOM, which was more
pronounced for the perennial than the seasonal vegetation (Figure 5b).

The simulated responses to changes in precipitation ($P$) at six Australian catchments revealed that the re-optimized VOM

followed a Budyko curve most closely (i.e. lowest values of the mean squared error, Figure 6) compared with the conceptual
hydrological models FLEX, TUWmodel and GR4J (see Supplement S3, Figures S3.1-S3.7 for time series of meteorological
data, model performances and resulting discharges). The hydrological models, with the same model parameters for each $P$
scenario, deviate from the Budyko curve in a similar way as the VOM without re-optimization (too flat around the unperturbed
$P$). When the hydrological models were run for a selection of CAMELS-catchments (see Supplement S3, Figures S3.8-S3.11)

their $n$-values also reduced for a 20% increase in precipitation and constant model parameters, but these changes remained
relatively small.



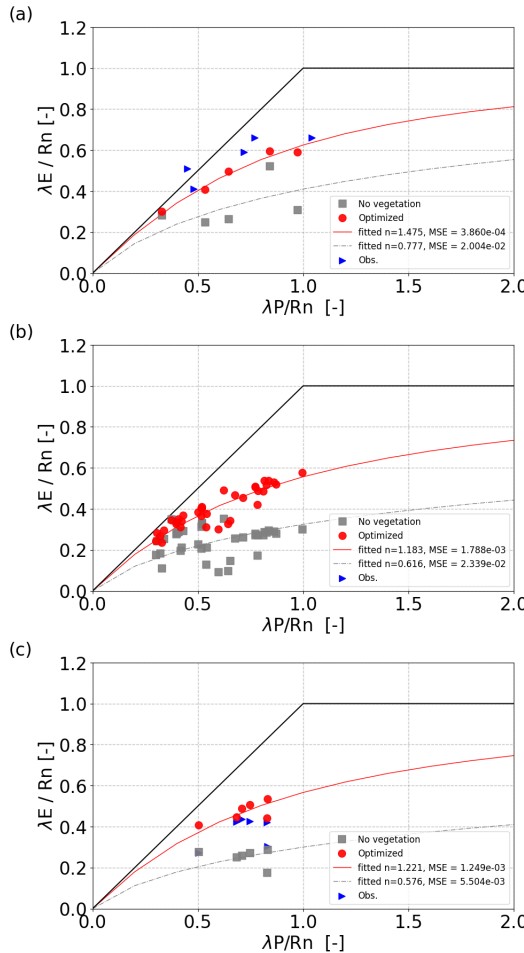

**Figure 2.** Results of the VOM in Budyko space, for a) flux tower sites along the North Australian Tropical Transect (NATT), b) 36 additional points around the NATT, c) six catchments around the NATT. The VOM with fully optimized vegetation is shown in red dots, the VOM run without vegetation (bare soil) in gray squares, observations (from flux tower data (a) or runoff data (c)) in blue triangles. Lines are fits of Eq. 3 to the data points of the same colour.

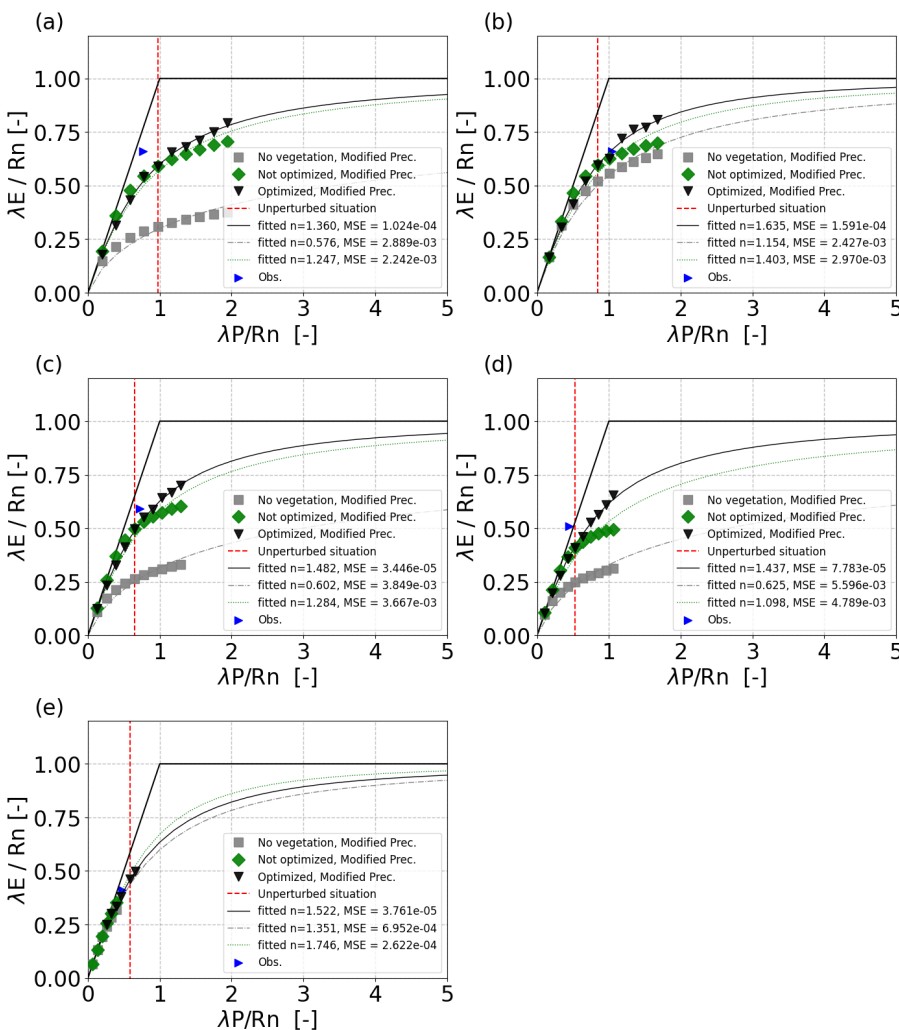

**Figure 3.** VOM-results along the NATT in Budyko space as a result of modified precipitation for a) Howard Springs, b) Adelaide River, c) Daly Uncleared, d) Dry River and e) Sturt Plains. Gray squares denote simulations without vegetation (bare soil), black triangles denote fully optimized simulations, while green diamonds denote simulations where the VOM was run with the optimal vegetation properties determined for an unmodified climate. The unmodified climate is indicated by the dashed red line. The blue triangles denote the eddy covariance observations, summed over the available period for each site. Lines are fits of Eq. 3 to the data points of the same colour.





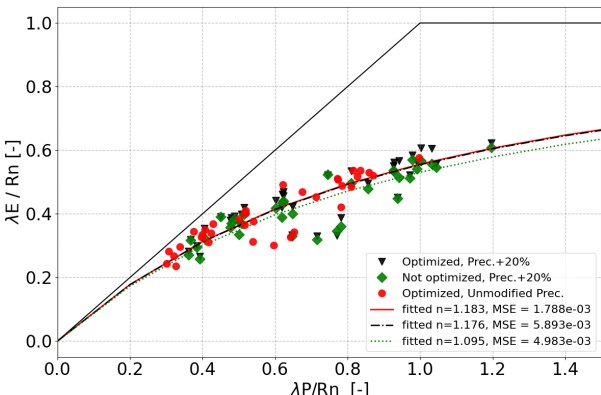

**Figure 4.** VOM-results along the NATT in Budyko space as a result of modified precipitation for 36 additional locations. Black triangles denote fully optimized simulations with modified precipitation, while green diamonds denote simulations where the VOM was run with the optimal vegetation properties determined for an unmodified climate (red dots). Lines are fits of Eq. 3 to the data points of the same colour.

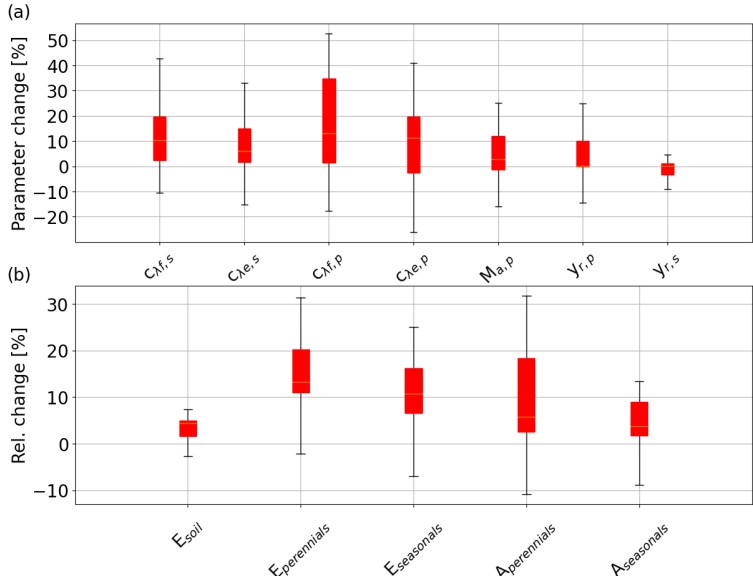

**Figure 5.** Simulated changes in response to a 20% increase in precipitation in the VOM (based on Fig. 3) for (a) optimized vegetation properties, with the water use parameters for perennial ($c_{\lambda f,p}$ and $c_{\lambda e,p}$) and seasonal vegetation ($c_{\lambda f,s}$ and $c_{\lambda e,s}$), perennial vegetation cover ($M_{A,p}$) and root depths for the perennial ans seasonal vegetation ($y_{r,p}$ and $y_{r,s}$) and (b) fluxes with soil evaporation ($E_{soil}$), transpiration of perennial trees ($E_{perennials}$), transpiration of seasonal grasses ($E_{seasonals}$), gross primary productivity (GPP) of perennial trees ($A_{perennials}$) and GPP of seasonal grasses ($A_{seasonals}$).



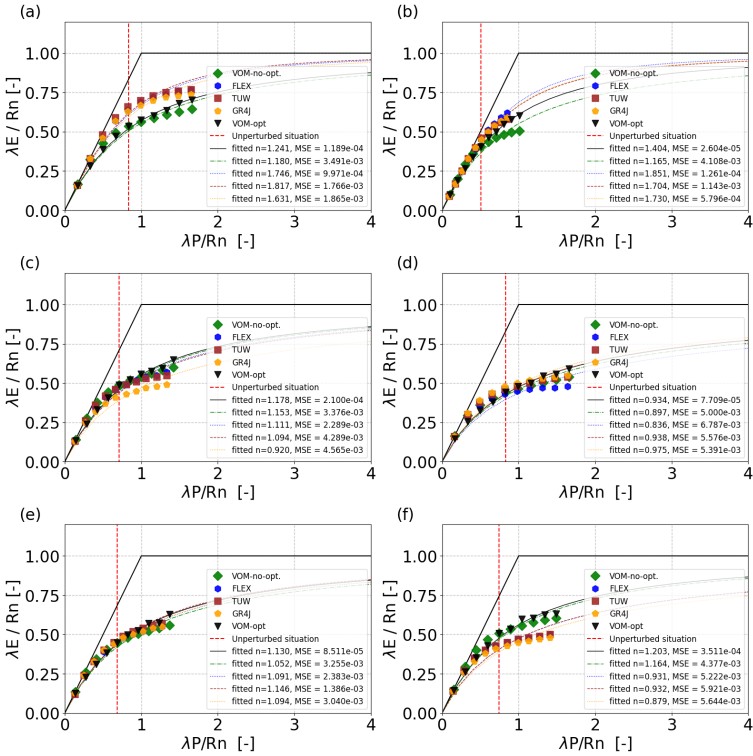

**Figure 6.** Simulations in Budyko space for six catchments in Australia as a result of modified precipitation for the optimized VOM (black stars), VOM without optimization (green triangles), FLEX (red diamonds), TUWmodel (gray dots), and GR4J (gold squares) for a) Adelaide River, b) Dry River, c) Fergusson River, d) Magela Creek, e) Seventeen Mile Creek and f) South Alligator River. The unmodified climate is indicated by the dashed red line.

### 3.3 Sensitivity of site-specific $n$-values to changing precipitation

In the next step, the $n$-values were treated as site-specific properties, and the sites and catchments were each fitted to Eq. 3 individually and per precipitation ($P$) scenario (i.e. a separate $n$-value for each data point in Figures 3 and 6). The $n$-values resulting from re-optimized long-term vegetation properties for each $P$ scenario were considerably less variable at each site than those obtained for constant long-term vegetation properties, both at each NATT site (Figure 7), as well as across the 36 additional sites along the NATT (Figure 9a). For these sites, increasing precipitation by 20% without re-optimizing the vegetation resulted in a reduction in $n$-values by around 0.10 (blue box, Figure 9a), whereas the re-optimized VOM-simulations (red box, Figure 9a) did not result in a systematic change in $n$. Surprisingly, for re-optimized vegetation, the variation in $n$ between sites was greater than at each individual site, even though $P$ was varied by an order of magnitude (Figures 7 and 8a). The opposite was the case if the long-term vegetation properties were not re-optimized for each perturbed $P$ (Figures 7 and 8b), or if the conceptual hydrological models with constant parameters were used (Figure 8c-e). In all these latter model simulations, the $n$-values systematically declined with positive perturbations in $P$ and increased with negative $P$ perturbations.



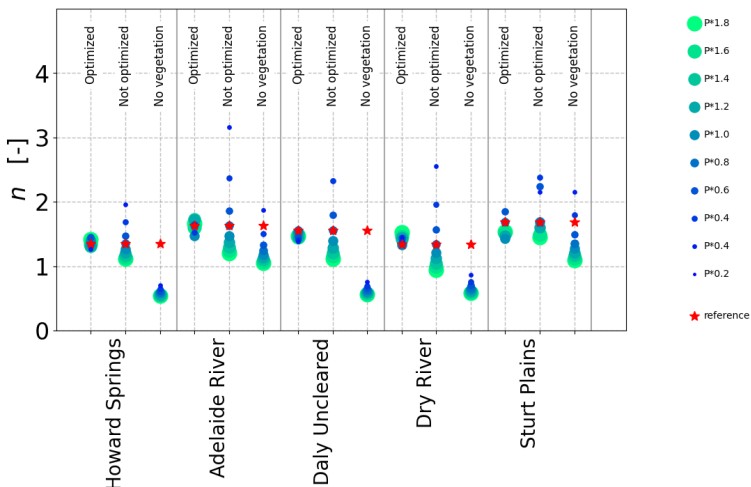

**Figure 7.** Variability of $n$ in response to changing precipitation in the VOM. A separate $n$-value was fitted to each point in Fig. 3. Sizes of dots represent the multiplication factor by which precipitation was modified at each site, while the red stars indicate simulations with unmodified precipitation. "Optimized": fully optimized vegetation; "Not optimized": optimal vegetation properties of unmodified precipitation; "No vegetation": bare soil simulations.

In an additional analysis, the three hydrological models were applied to a selection of catchments of the CAMELS data, to prove the generality of our findings. Interestingly, in this analysis we found that the sensitivity of $n$ to changes in $P$ was relatively similar (median decrease in $n$ by 0.1 in response to a 20% increase in $P$) compared with the non-optimized VOM
(Figure 9a), although the models were run over very different sets of conditions, i.e. the CAMELS catchments in the US for the hydrological models and sites along the NATT in Australia for the VOM (see also Supplement S3, Figures S3.8-S3.11 for all results).

## 4  Discussion

The results of our study provided new insights into the likely principles underlying convergence of catchments along the
10 Budyko curve and shed new light into the expected sensitivity of vegetation and the catchment water balance to changes in climate. Here we systematically discuss the questions raised and hypotheses formulated in the introduction, before proceeding to more general observations and limitations of this and similar studies.

### 4.1  Reproduction of the Budyko curve by optimality

The first hypothesis formulated in the introduction (H1) is that convergence of catchments along the Budyko curve, i.e. close
to the maximum possible long-term evapo-transpiration, results from vegetation optimality:



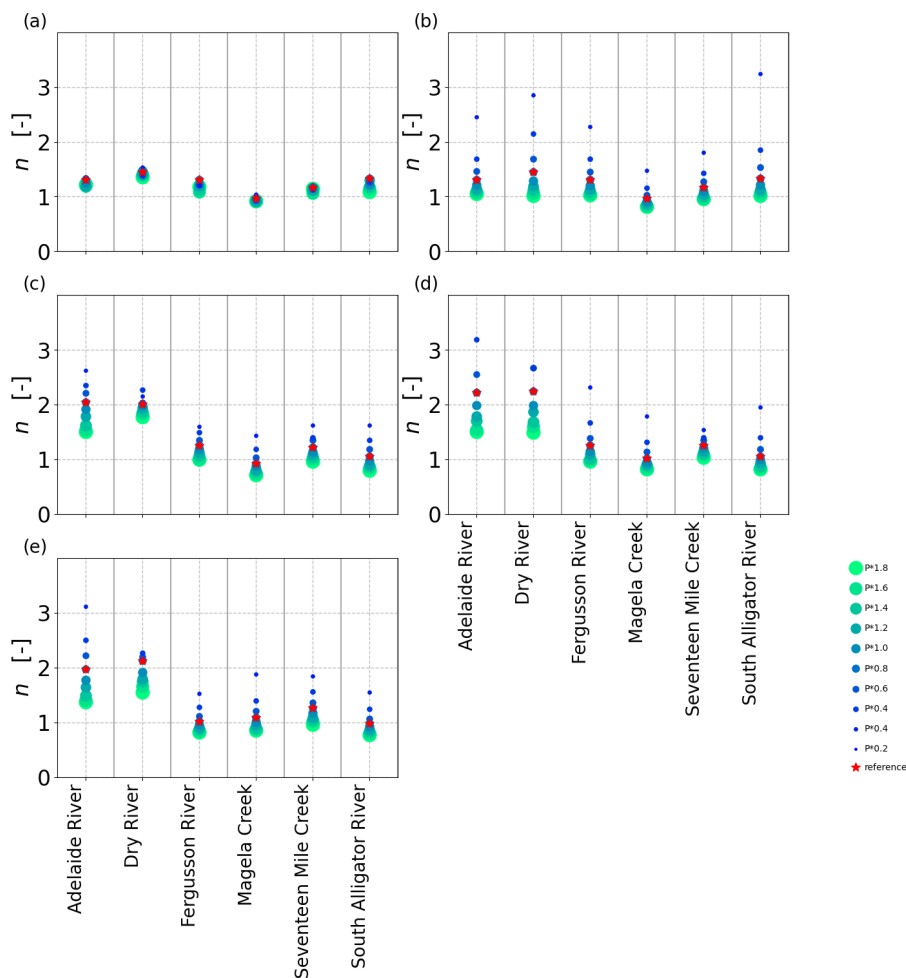

**Figure 8.** Variability of $n$ in response to changing precipitation in a) the optimized VOM, b) the VOM without re-optimization, c) FLEX, d) TUWmodel, and e) GR4J. Fitted $n$-values for six catchments in Australia with unperturbed precipitation (red star) and increased/decreased precipitation (colour scale).





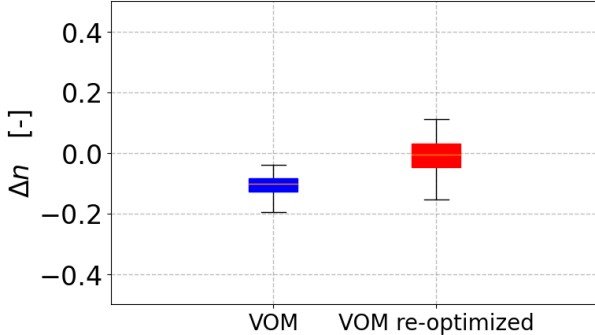

**Figure 9.** Changes in $n$ in response to a 20% increase in precipitation ($P$). VOM simulations are shown in for 36 sites along the NATT, with the blue box (left) representing VOM simulations where the long-term vegetation properties were not re-optimized to the increased $P$ and the red box (right) representing simulations where all vegetation properties were re-optimized for the increased $P$. $n$ values were obtained by fitting Eq. 3 to each individual site for a given $P$ scenario.

H1 Model simulations based on vegetation optimality lead to a better reproduction of the empirical Budyko curve than model simulations without self-optimized vegetation.

This hypothesis is clearly supported by the following findings: (a) The VOM with fully optimized vegetation follows more closely Budyko curves with realistic $n$-values (1.1-1.5) than the VOM without vegetation (Figure 2). In comparison, literature $n$-values often range between 1.5 and 2.6 (e.g. Roderick and Farquhar, 2011; Choudhury, 1999; Yang et al., 2008). Note that the values found here are lower than the recommended value of 1.8 by Choudhury (1999) or the value of 2.0 used in the Turc-Pike relationship (Pike, 1964), but compare well with the $n$-value of 1.49 found by Williams et al. (2012) for 176 flux tower sites. In contrast, the $n$-values of 0.62-0.78 produced by the VOM simulations without vegetation (bare soil evaporation only) are well below any values reported in the literature. (b) Simulations with perturbed precipitation ($P$) revealed that the VOM with re-optimized vegetation for each $P$ setting followed the same Budyko curve very closely, whereas none of the other models did (Figures 3, 4 and 6). This happened consistently for different study sites and will be discussed in more detail in the next section.

Altogether, these results suggest that vegetation optimality has a strong tendency to push the catchment water balance closer towards the envelope compared to simulations without vegetation, but also to keep it on a catchment-specific Budyko curve as climate changes ($n$ varied more between catchments than within each catchment as $P$ was varied by an order of magnitude in Figure 7). So far, the Budyko framework was related to optimality theory only a few times, mostly by applying thermodynamic optimality principles and constraints in numerical experiments (Porada et al., 2011; Kleidon et al., 2014; Westhoff et al., 2016; Wang et al., 2015). To our knowledge, the results presented here illustrate for the first time that an ecologically motivated optimality principle (maximum net carbon profit) leads to a close reproduction of the Budyko curve. Previously, Milly (1994) suggested that convergence towards the Budyko curve may be a result of plants optimizing their rooting depths to maximize





transpiration in a given environment. However, here we found that convergence on the Budyko curve is likely the result of rooting depths, vegetation cover and water use strategies playing together in a way to satisfy a biological optimality principle.

## 4.2 Sensitivity of $n$-values to changes in precipitation

Our second hypothesis (H2) reflects a common belief expressed in the literature whenever the Budyko framework is used to study the effects of climate change:

H2 The empirical parameter $n$ stays constant as climate changes, as long as vegetation cover and rooting depths stay constant.

Based on our results for varying precipitation ($P$), this hypothesis has to be clearly rejected. Constant long-term vegetation properties in the VOM and constant parameters in the conceptual hydrological models led to a large spread in $n$-values, with a systematic decline in $n$ for increasing $P$ and increase in $n$ for reduced $P$ (Figures 7, 8). In our study, only full optimization of vegetation properties to changing precipitation led to a constant $n$ as $P$ was changed step-wise, by up to an order of magnitude (Figures 7 and 8a). But this optimization, in fact, represents a change in vegetation properties as $P$ changes, disproving the

above hypothesis in the context of the four models applied in the present study.

Our findings bring forward one of the main deficiencies that many conceptual hydrological models currently have, namely the missing ability to adjust system properties (i.e. parameters) in response to environmental change, an ability that would be needed for predictions under change (Montanari et al., 2013; Ehret et al., 2014). Calibrating conceptual model parameters to past observations and then using the calibrated models for prediction under environmental change implicitly assumes that the

model parameters represent static catchment properties, which should not change as e.g. the climate changes. The flaw in this assumption becomes immediately clear if one considers that some of these parameters encode vegetation functioning, which cannot be assumed static as the environment changes. The VOM and other process-based models offer a clearer separation between parameters related to physical catchment properties and those encoding vegetation functioning. Especially the optimality part enabled the VOM to track the Budyko curve under changing $P$, which is very encouraging for the implementation

of vegetation optimality for predictions under change, as already shown for predicting vegetation response to elevated $CO_2$ (Schymanski et al., 2015).

## 4.3 Long-term vegetation response and resulting $n$-values

The last hypothesis relates to the underlying reasons for a change in $n$-values:

H3 Changes in $n$-values are a result of slowly varying, long-term vegetation properties.

As already hinted at in the previous section, this hypothesis has to be rejected, too, as our results suggest the opposite. Only if the slowly varying, long-term vegetation properties were re-optimized in response to changing precipitation ($P$) did $n$ stay constant, otherwise, $n$ changed in the opposite direction to $P$ (Figure 5). This seems in line with the long-term character of





the Budyko curve itself, but actually conflicts with findings that variations around the curve correlate with variations of the long-term average annual vegetation coverage (Li et al., 2013) or forest cover (Shao et al., 2012). Interestingly, the simulated changes in rooting depths in response to increased $P$ (Figure 5) agree with the general observation that rooting depths increase with increasing precipitation amounts (Schenk and Jackson, 2002). In a previous study, we found that the VOM reproduces

this pattern along a precipitation gradient (Nijzink et al., 2022a), whereas here we demonstrate that the VOM also maintains this pattern over time for constant physical catchment properties.

Our finding that $n$ varied less in response to changes in $P$ than between catchments (Figure 7) suggests that the $n$-values are indeed site-specific. However, our analysis did not reveal whether $n$ depends on properties of climate (e.g. temperature, seasonality of net radiation and precipitation, rainfall intensity), or physical catchment properties (e.g. soil water holding

capacity), or both. Previous studies also suggested that $n$ is a catchment-specific variable depending on climate properties (other than mean annual $P$ and $R_n$), physical catchment properties and/or vegetation (Donohue et al., 2007; Oudin et al., 2008; Yang et al., 2009; Williams et al., 2012).

However, our findings challenge the common belief formulated in H3 that changes in $n$ are a result of changes in the physical catchment properties or vegetation, whereas $n$ should stay constant if only mean annual $P$ or $R_n$ change (Roderick

and Farquhar, 2011; Renner et al., 2012). Our results suggest quite the opposite, as changes in the long-term, slowly varying vegetation properties were needed to bring a certain catchment back to the original $n$-value after a change in $P$. Since vegetation reacts to the climate, the parameter $n$ can be considered a combined result of climate, vegetation and landscape properties (Zhang et al., 2004; Roderick and Farquhar, 2011; Donohue et al., 2012; Ning et al., 2017), but interestingly, vegetation optimality suggests that the vegetation response to changes in climate stabilizes $n$. Our results also emphasize that the dynamic

evolution of the system is important, as argued before by e.g. Koster and Suarez (1999), Yang et al. (2007). If a climatic shift happens quickly, and the state of the system did not yet adapt to the new situation, we may expect different $n$-values than for a system in a (dynamic) steady-state. Note that other authors were not able to find that catchments followed a single Budyko trajectory over time (Reaver et al., 2020).

## 4.4   Dryness and Wetness Index

Different ways to plot the Budyko framework have been used in the literature, either using a dryness index ($R_n/\lambda P$) or a wetness index ($\lambda P/R_n$) as the independent variable. For this reason, we repeated our analyses in Supplement S4 for different projections of the Budyko framework. The previous results in Sections 3.1-3.2 remained valid after using a Budyko space based on a dryness index (Figures S4.1-S4.5 in Supplement S4). Nevertheless, it must be noted that the resulting $n$-values differ for a Budyko space with a wetness index instead of a dryness index. More specifically, Figure S4.5 shows, for example,

that the resulting $n$-values for the CAMELS data slightly differ. However, the $n$-values still similarly change for a Budyko-plot with the dryness index $\lambda P/R_n$ as the independent variable, as for a Budyko-plot with a wetness index (Supplement S3, Figure S3.8).

Note also that the results in Sect. 3.3 would not change for a different Budyko-projection, as these $n$-values could be solved analytically. However, when using a dryness index (i.e. plotting $E/P$ as a function of $R_n/\lambda P$) the interpretation of the results





becomes more difficult, as e.g. an increased $P$ would decrease $E/P$ at constant $E$. Therefore, changes in $E$ do not directly translate to changes in $E/P$. For this reason, we used a wetness index in the main paper (i.e. plotting $\lambda E/R_n$ as a function of

$\lambda P/R_n$), in which case a change in $P$ only results in a change on the vertical axis if $E$ changes.

## 4.5   Limitations

In the present study, we perturbed precipitation in isolation, i.e. without perturbing any other climate variables. However, changes in precipitation are in reality linked to changes in vapour pressure, net radiation (e.g. through cloud formation) and likely also different rainfall intensities. We also attempted to vary atmospheric water demand, but were limited by the differ-

ences of the VOM and conceptual hydrological models. The hydrological models use potential evaporation as input, whereas the VOM uses vapour pressure, temperature and incoming shortwave radiation to drive evaporation. VOM simulations with altered vapour pressure and incoming shortwave radiation are presented in Supplement S2. Especially when altering the vapour pressure, results became difficult to interpret, as net longwave radiation and hence the position along the x-axis of the Budyko plots is only indirectly affected by vapour pressure (Equation 39, Allen et al., 1998). Here, the results strongly depended on the

projection in Budyko space, i.e. whether a dryness or wetness index was used (see Supplement S2, Figure S2.3). In contrast, changes in shortwave radiation confirmed that vegetation optimality reduces departures from the Budyko curve compared to static vegetation (Supplement S2, Figures S2.1 and S2.2), but in a less systematic way than precipitation. This related to short-wave radiation being a strong driver for photosynthesis in the VOM, which profoundly affects all components of vegetation.

Note that this study did not aim to predict responses of vegetation and the water balance to any realistic climate change

scenario, and that the finding of a stabilizing effect of vegetation optimality on the value of $n$ as precipitation changes may not apply to human-induced climate change at all. This is most obvious when considering that rising atmospheric $CO_2$ concentrations are the main driver of human-induced climate change, as they have a profound, direct effect on vegetation water use (which can be buffered, enhanced or even reversed by vegetation optimality, depending on the climate), which would modify a site-specific $n$ value permanently, even in the absence of any other climatic changes (Schymanski et al., 2015).

## 25  5   Conclusions

The Vegetation Optimality Model showed a strong convergence to the empirical Budyko curve for six eddy covariance sites, 36 additional locations along the North Australian Tropical Transect as well as six catchments in Australia, if vegetation was re-optimized for perturbations in precipitation. In contrast, the VOM with constant vegetation properties, as well as three hydrological models with constant parameters, led to systematic changes of the parameter $n$ defining the curve, with increased

$n$-values for decreased precipitation, and decreased $n$-values for increased precipitation.

Our study results have a range of potentially important implications. First of all, the finding that vegetation optimality explains convergence of catchments on the empirical Budyko curve lends support to further explore optimality principles for the prediction of vegetation water use in ungauged situations and in a changing environment. Secondly, if the vegetation response to changes in precipitation ($P$) indeed follows the predictions of the VOM, we may use the Budyko framework for



predicting sensitivities of the catchment water balance to $P$ as was done in several previous studies. However, our results motivate an expectation that $n$ should be conserved at a given catchment in the long term, whereas it may vary in the short to medium term as climate changes. This highly contrasts with previous assumptions that the parameter $n$ should stay constant in the short term as climate changes, and then vary in the longer term. Nevertheless, the long-term conservation of $n$ would make the Budyko framework even more useful, as long-term predictions are notoriously difficult.

Furthermore, we found that vegetation adaptation to climatic change may not always lead to an increase in transpiration, but also to down-regulation of vegetation properties, depending on the type and direction of climate change. For example, in our study, the VOM and all other models predicted that increases in $P$ would reduce $n$ while decreases in $P$ would increase $n$, if vegetation does not fully adapt to the new $P$. This means that in a climate change scenario with reduced $P$, vegetation water use may keep decreasing slowly as vegetation re-adjusts its rooting depth, cover and water use strategies to the new setting, bringing $n$ back to its original value.

However, climate change never affects only one aspect of atmospheric forcing, and given that e.g. rainfall intensity and seasonality of water and energy availability has an impact on $n$, the assumption of constant $n$ as climate changes may not be adequate. Moreover, one important aspect of climate change has not been touched upon at all in this study, which is the effect of rising atmospheric $CO_2$ concentrations, which may have a lasting effect on $n$ all by themselves, on top of their effect on local climate. For that reason, we advocate the inclusion of $CO_2$ concentrations in any analysis of climate change effects on vegetation water use. Nevertheless, our finding that vegetation optimality tends to keep $n$ close to a site-specific value serves as a strong motivation to investigate further what aspects of climate and physical catchment properties most likely affect this value, as such knowledge may enable robust predictions about changes to the water balance in response to climate change.

*Code and data availability.* Model code is available on github (https://github.com/schymans/VOM) and the full analysis including all scripts and data are available on renku (https://renkulab.io/projects/remko.nijzink/budyko).

*Author contributions.* SJS and RCN designed the study. Simulations and analyses were carried out by RCN. SJS and RCN contributed to the final text.

*Competing interests.* On of the authors is a member of the editorial board of Hydrology and Earth System Sciences.

*Acknowledgements.* This study is part of the WAVE-project funded by the Luxembourg National Research Fund (FNR) ATTRACT programme (A16/SR/11254288).



We acknowledge the TERN-OzFlux facility (http://data.ozflux.org.au/portal/home, last access: 9 February 2022). OzFlux was supported financially by the Australian Federal Government via the National Collaborative Research Infrastructure Scheme and the Education Investment Fund.

We thank the SILO Data Drill hosted by the Queensland Department of Environment and Science for providing the meteorological data (https://www.longpaddock.qld.gov.au/silo/, last access: 22 October 2020).

We acknowledge the Northern Territory Water Data WebPortal for the river discharge data (https://water.nt.gov.au/, last access: 10 February 2022).

We thank the Scripps $CO_2$ programme (https://scrippsco2.ucsd.edu/data/atmospheric_co2/primary_mlo_co2_record.html, last access: 10 January 2020) for the Mauna Loa Observatory records.

We thank NCAR for making the CAMELS data available ( https://ral.ucar.edu/solutions/products/camels , last access: 9 February 2022).

We also thank CSIRO for the Soil and Landscape Grid of Australia (https://www.clw.csiro.au/aclep/soilandlandscapegrid/, last access: 1 April 2020).





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
