# Peer review of "Vegetation optimality explains the convergence of catchments on the Budyko curve"

_Hydrology and Earth System Sciences, 2022_

## Author Response (AR1)

**Final response**

We would like to thank the referees and the editor for their constructive feedback and comments, which we used to revise our manuscript. In the below, we provide a list of changes in the manuscript. In addition, we addressed the remaining questions of the editor and added and updated our responses to the referee comments. The editor and referee comments are written in italics, line numbers in our responses refer to lines in the revised manuscript.

**List of changes**

- New figure 2 with the experimental design.

- More extensive description of the experimental design.

- Extra equations of the VOM related to transpiration.

- New Supplement S6 with more model details.

- Addition of Table 3 with calibrated model parameters of the hydrological models.

- Textual changes in Discussion and Limitations based on the comments.

**Response to Editor Comments**

*I read your paper, the reviewer comments and your responses with great interest. In line with both reviewers, I think your study addresses a key question in eco-hydrology. Yet, both reviewers criticize that the experimental design is hard to follow, the theory and equations underlying VOM should be better explained and that the link to the other three (why three?) conceptual models is not obvious. In conclusion and in line with both reviewer I think that your study needs thorough revisions to become acceptable. In this respect, I'd like to stress that the changes you outlined in your responses are very much appropriate.*

Thank you for assessing the manuscript. We improved the description of the experimental design, with a flow chart and extra explanations. We added detailed equations and descriptions of the VOM in Supplement S6 as well as in the Methods section.

*Additionally, to the recommendations raised by the reviewers, I take the liberty to add a few thoughts that might be helpful in for improving the manuscript.*

*- The way climate change is simulated is a crude simplification, as it ignores land surface atmosphere feedbacks. I would assume a shift in the Bowen ratio to larger values and thus stronger sensible heat fluxes, higher air temperature as well as warmer and dryer soils when reducing precipitation to 20%. Also the water vapor pressure deficit will change. Were all these aspects ignored and the meteorological drivers kept constant during these scenarios?*

We are aware that simulating climate change in this way is a crude simplification. Indeed, these aspects were ignored and the other meteorological drivers were kept constant. We discussed this already in Section 4.5 in the manuscript, but we added now a mention of the expected shifts as described above.

*- The results that VOM without vegetation has strongly lowered evaporative fraction is very interesting (although not too surprising). Does this imply that the Budyko curve is only valid for vegetated catchments?*

Good point. the following text was added in Section 4.1:

"Many other authors have shown previously that the evaporative fraction decreases due to land clearing. For example, Li et al. (2013) found a positive relation between the curve shape parameter and vegetation cover, implying an increasing evaporative fraction with increasing vegetation cover. As discussed below, our results suggest on top of this that for a changing climate, this shape parameter would only be conserved if vegetation is free to co-evolve with the climate, which would not be the case in a cleared catchment."

*- The estimate of PET based on Rn/lambda is in fact the energy limit but not PET. While I am aware that Budyko used this estimate as well, it is still not correct. And, I expect that the three conceptual models use estimated of PET as well. So is this consistent?*

Thank you for this remark, it is indeed the case that the hydrological models use the potential evaporation from the SILO DataDrill, which was determined with the Penman formula following the FAO (Allen et al., 1998). In the original manuscript, we used PET instead of Rn in the Budyko plots,

which is indeed inconsistent with the Rn-based calculations for the VOM results. This is now corrected in the revised manuscript, where all positions in Budyko space are calculated using Rn, but the overall results did not change significantly. We do not believe that the use of PET should be preferred over Rn in the Budyko framework, as PET is usually calculated using atmospheric vapour pressure, temperature and sometimes even wind speed, but for large catchments these should be considered internal variables due to surface-atmosphere feedbacks. Even Rn, which is a function of the surface temperature, is strictly speaking not an external forcing, but this discussion is beyond the scope of this paper, and therefore we used Rn, as in Budyko's original proposition.

*- The Budyko theory is valid for large catchments, but VOM is a 1 d model. Does this imply that the authors believe that a single column is sufficient for simulating a catchment of many 100 sqkm?*

The reasons why the Budyko framework works best for large catchments are likely related to reduced effects of below-ground advection and small-scale variations of land use. Here we simulate much smaller catchments, or in the case of the VOM, actually idealised blocks of soil with vegetation on top. Neither the VOM nor the hydrological models employed here have any lateral dimension, so their range of validity in terms of catchment size is undefined. The only observations used in comparison with the VOM simulations here stem from eddy covariance measurements, which represent a fetch of a few hundred meters, not hundreds of square kilometers. But your comment made us realise that the reader may be misled to believe that the VOM is applicable at the catchment scale, so we added more discussion about this in Section 4.5.

*- I found your response to the representation of vegetation in the conceptual models not really convincing. As a modeler you have free choice to implement a more complex approach for ET in the models, that accounts for LAI and rooting depth and their seasonal dynamics.*

Of course, there are more complex approaches available to model ET, which are also likely better suitable for modelling change. The models we used here are, however, relatively simple and some are still intensively used in catchment hydrology. These simple models were chosen to test the robustness of our finding that changing precipitation leads to changing n if vegetation does not adapt. The changing n in these simple models without explicit vegetation component illustrates most convincingly that the change is not brought about by vegetation processes, as suggested in the literature previously. We clarified this in Section 4.5.

*- Maybe it is a misinterpretation – but none of the observations (the blue triangles) did actually drop on the fitted Budyko curve, or am I wrong? So is this an indicator that VOM has a structural error? Or is the vegetation in the catchment is not in an optimum?*

Thank you for pointing this out. In Figure 4, two of the blue triangles fall on the Budyko curve, one misses it by a mile, and one is even outside of the Budyko envelope. This has no meaning for the adequacy of the VOM, but due to the fact that the observations relate to much shorter time series than those used for the simulations. The Budyko framework applies to very large catchments where lateral advection can be ignored and to very long time series where changes in storage become negligible compared to the cumulative fluxes. We clarify this in the discussion, Section 4.5.

**Response to Referee #1**

We would like to thank Referee #1 for the assessment of our manuscript and the constructive feedback. The referee raised several issues, which we address below with the referee comments written in italics.

*This manuscript used three conceptual hydrological models to model evaporation under different vegetation conditions. In the current version, it isn't clear how to deal with the impact of changing vegetation in the hydrological models. More descriptions on the parameterization of vegetation, including root depth, are required.*

These conceptual models generally use a few parameters that can be linked to vegetation. In general, they use one or several parameters that determine the storage in the root zone, which is related to rooting depths, although not being exactly the same. At the same time, a threshold soil moisture is usually defined after which the transpiration occurs at a potential rate. These parameters are all calibrated for the conceptual hydrological models, for the different cases in our experiments. Hence, changes in vegetation are represented by different values for the vegetation-related parameters, obtained by calibration.

We added a more extensive description per model which parameters are related to vegetation. In addition, we added a table with the vegetation-related model parameters and a description. We added Supplement S6 with more model details.

*I suggest a flow chart for the experimental design, which can help readers more easily understand the design scheme.*

Thank you for this idea, we added a flow chart with the experimental design in Figure 2.

*It is interesting that n might initially increase in response to suddenly reduced P, and only slowly returns to into original. It is valuable to exhibit the evolution process that the parameter n slowly returns to its original, and how long it can return to its original value. In a previous study, Zhang et al. (2016, GRL, doi:10.1002/2015GL066952) found a linear relationship of n with vegetation during 1982-2011, which possibly indicates that n can't return to its original for a 30-years period.*

Thank you for this valuable remark and the reference. We added this reference to our discussion.

*I agree with the authors that vegetation is a result of climate. Yang et al. (2014, JoH, http://dx.doi.org/10.1016/j.jhydrol.2014.05.062) found that the parameter n has a logarithmic relationship with catchment slope, but doesn't have a significant relationship with vegetation coverage. I guess that vegetation is the result of climate and catchment characteristics (such as slope, topography, permeability and etc.), and consequently the relationship of n with vegetation can be contained in the relationship of n with catchment characteristics.*

Thank you for this additional confirmation of our finding that vegetation stabilizes the n-value for a given catchment by its adaptation to the catchment properties. We added this to our discussion.

**Response to Referee #2**

We would like to thank Referee #2 for the constructive comments on our manuscript. Below, we address the comments, with the referee comments written in italics.

*Why did the authors use rooting depth in the vegetation optimality model? Since the conceptual models, e.g. FLEX, used the root zone storage capacity as the term in modeling. What is the relationship between the rooting depth and the root zone storage capacity? Since root zone storage capacity is a more meaningful term to describe the interaction between ecosystems and hydrology in catchment scale, like previous studies (https://agupubs.onlinelibrary.wiley.com/doi/full/10.1002/2014GL061668; https://hess.copernicus.org/articles/20/1459/2016/hess-20-1459-2016.pdf ), rather than their depth. I'd like to hear the authors' thoughts. And this part needs more clarification.*

Thank you for bringing this up, we will add more discussion in the revised manuscript about the differences between the models and their parameters. Indeed, these models are extremely different in their concepts, but this makes it only more interesting that we found similar results. In the VOM, the plants are optimized for maximum net carbon profit, by adjusting vegetation properties such as as rooting depths and vegetation cover. As a result, this leads to an effective root zone storage capacity, which is not explicitly quantified. The conceptual models lump vegetation and catchment properties directly in effective model parameters such as the root zone storage capacity. Since the parameters in the conceptual models were calibrated, we did not undertake a comparison between the effective root zone storage capacities. However, we will point the reader to alternative approaches of estimating root zone storage capacities, also based on the given references.

*There are lack of equations described the vegetation model. For readers who did not have enough background about this model, it is very necessary to give some equations, illustrating how the model describes the core processes. Adding a table showing the equations might be a good idea.*

Thank you for this idea, we added an extensive description in Supplement S6 (i.e. more than the table we mentioned in our initial response) with the main equations of the VOM. We added in Section 2.3.1 the equations of the VOM related to transpiration and the Net Carbon Profit.

*I found the experiment design are hard to follow. And the VOM and three hydrological models are not well linked. A workflow chart might be helpful.*

Thank you for this suggestion, we added a scheme with the experimental design as also requested by Referee #1.

*The Budyko equations are not the same as we widely used. This brings in much difficulty to review this manuscript. I did not see a clear reason to use Rn rather than potential evaporation (Ep) in the traditional Budyko equation.*

Please note that the original Budyko framework (Budyko, 1974) was also formulated in terms of net radiation, which we adopted here as well. We also decided to use Rn, as this is a more general form of defining the energy for evaporation, which is consistently used by the different models. For example, the VOM does not use potential evaporation in terms of water (i.e. in mm/day), but only downwelling

radiation for photosynthesis. Also the form of the equation is rather common, it was originally defined by Mezentsev (1955) and later, among others, Choudhury (1999) and Roderick and Farquhar (2011).

*The authors conducted the study for 37 years (1-1-1980 until 31-12-2017). I am curious to know whether there were any changes during these 37 years. This needs more clarification and discussion.*

Thank you for this comment. The flux tower sites are relatively undisturbed, but only suffer occasionally from fires. We will mention recently documented trends at Howard Springs over time in the revised manuscript, see also Hutley et al. (2022). For the catchments and additional locations along the transect, we can, unfortunately, not state with certainty that there were no changes. However, since we only use the full time series of the climate data as input for a numerical experiment, any trends in the fluxes would not make a difference to our results.

We added a paragraph in Section 4.5 Limitations about this.

*In Section 2.3.2, what does this mean "a permeable soil block with layers of 0.2 m thickness, and a total thickness of 30 m"?*

The VOM schematizes the soil as a block with layers of 0.2 m. In total, there are 150 layers, leading to a soil depth (to impermeable bedrock) of 30 m. Vertical flow between the layers is possible down to the last impermeable layer. Lateral drainage is simulated from the bottom saturated layers. We clarified this in the manuscript.

*The equations used in TUWmodel and FLEX to estimate evaporation are the same, although with a bit different term definition. That is why all three conceptual models have very similar performance. It seems not necessary to use three conceptual models in this study.*

Indeed, these models have quite some similarities, but also differ at several points. Especially the GR4J-model has a rather different formulation of the actual evaporation (Equation 6), but Equation 5 and 7 of the TUW and FLEX model are basically the same. Nevertheless, these models still differ in other aspects that result in different outflows, storages and a resulting evaporation. The configuration of the conceptual reservoirs is different, as well as the definitions of fast flow components and the simulation of interception evaporation.
We decided to use multiple models here, in order to assess the robustness of our results. We added more discussion and explanation about the differences and similarities of the models, as well as the robustness of our findings.